# Water-Retaining Polymer and Planting Pit Size on Chlorophyll Index, Gas Exchange and Yield of Sour Passion Fruit with Deficit Irrigation

**DOI:** 10.3390/plants13020235

**Published:** 2024-01-15

**Authors:** Antônio Gustavo de Luna Souto, Edinete Nunes de Melo, Lourival Ferreira Cavalcante, Ana Paula Pereira do Nascimento, Ítalo Herbert Lucena Cavalcante, Geovani Soares de Lima, Rafael Oliveira Batista, Hans Raj Gheyi, Reynaldo Teodoro de Fátima, Evandro Franklin de Mesquita, Gleyse Lopes Fernandes de Souza, Guilherme Romão Silva, Daniel Valadão Silva, Francisco de Oliveira Mesquita, Palloma Vitória Carlos de Oliveira

**Affiliations:** 1Postgraduate Program in Soil and Water Management, Universidade Federal do Semiárido, Mossoró 59625900, RN, Brazil; rafaelbatista@ufersa.edu.br (R.O.B.); daniel.valadao@ufersa.edu.br (D.V.S.); palloma.oliveira@alunos.ufersa.edu.br (P.V.C.d.O.); 2Postgraduate Program in Agronomy, Universidade Federal da Paraíba, Areia 58397-000, PB, Brazil; enm@academico.ufpb.br (E.N.d.M.); lofeca@cca.ufpb.br (L.F.C.); ana.nascimento@academico.ufpb.br (A.P.P.d.N.); italo.cavalcante@univasf.edu.br (Í.H.d.L.C.); gleyse.souza@academico.ufpb.br (G.L.F.d.S.); grs@aluno.ueg.br (G.R.S.); 3Postgraduate Program in Agricultural Engineering, Universidade Federal de Campina Grande, Campina Grande 58429900, PB, Brazil; geovani.soares@professor.ufcg.edu.br (G.S.d.L.); hans.gheyi@ufcg.edu.br (H.R.G.);; 4Department of Agrarian and Exact Sciences, State University of Paraíba, Catolé do Rocha 58429-500, PB, Brazil; evandrofranklin@servidor.uepb.edu.br; 5Postgraduate in Ecology and Conservation, State University of Paraiba, Campina Grande 58429-500, PB, Brazil; francisco.mesquita@servidor.uepb.edu.br

**Keywords:** *Passiflora edulis* Sims, water stress, pit volume, hydrogel, physiology, fruit harvest

## Abstract

Water availability is a limiting factor for the cultivation of sour passion fruit. Soil management techniques and the use of water-retaining polymers can increase soil water retention, reducing the frequency of irrigation in the crop. In this context, the objective of the research was to evaluate the gas exchange, the chlorophyll index, and the yield of the sour passion fruit cv. BRS GA1 as a function of irrigation depths, pit volumes, and doses of water-retaining polymer. The experiment was carried out in randomized blocks, in plots subdivided in a 2 × (2 × 5) arrangement, with irrigation depths of 70 and 100% of the crop evapotranspiration (ETc) as the main plot, the subplots with the volumes of pit of 64 and 128 dm^3^, and doses of the water-retaining polymer of 0, 0.5, 1.0, 1.5, and 2.0 g dm^−3^. The interaction of irrigation depths × pit volumes × doses of water-retaining polymer influences chlorophyll indexes, gas exchange, and water productivity, with positive impacts on yield of the sour passion fruit. The water depth of 70% of ETc increased the yield of sour passion fruit, in pits of 64 dm^3^. The application of doses of up to 1.1 g dm^−3^ of the water-retaining polymer and irrigation with water of 70% of ETc is recommended, and a dose of 2.0 g dm^−3^ of the water-retaining polymer in a pit volume of 128 dm^3^, associated with an irrigation depth of 100% ETc causes stress in sour passion fruit plants due to excess water.

## 1. Introduction

The sour passion fruit (*Passiflora edulis* Sims) is a species belonging to the Passifloraceae family, cultivated in tropical and subtropical countries, being one of the fruit species in greater evidence in Brazil [1]. In 2022, it had a national average yield of 15.26 t ha^−1^ [2]; however, this is still far below the genetic and productive potential of the crop, which can exceed 50 t ha^−1^ [3]. Brazil is the largest producer and consumer of passion fruit in the world, with the states of Bahia and Ceará as the largest producers of the species with 207,488 and 177,291 t, respectively [2].

Passiculture has great socioeconomic importance for the semiarid region of the Brazilian Northeast; however, water in this region is the most limiting factor for obtaining high yields, due to high evaporative rates and irregular distribution of rainfall, making it necessary to use irrigation [4,5]. However, the use of water by agricultural crops is considered very low, as approximately 60% of the applied water, depending on the irrigation method, is lost by evaporation [6].

The limitation of water availability in sour passion fruit can cause oxidative stress [7], negatively affecting several physiological processes, such as a reduction in chlorophyll levels [4], gas exchange [5], assimilation of carbon dioxide [8,9], and in fruit yield [4,10]. In this regard, the use of irrigation associated with techniques that allow for greater storage and retention of water in the soil close to the absorbent roots can increase irrigation efficiency with reflections on greater water availability to the plant.

One of these techniques is the use of water-retaining polymers in the soil, which can store and act as regulators of water availability for crops, providing a reduction in nutrient losses by leaching, increasing yield and minimizing production costs [11,12,13,14]. For the sour passion fruit, the use of the water-retaining polymer provided satisfactory results in the formation of seedlings [15,16], an increase in photosynthetic rates, and fruit yield [4,5]. In other fruit species, beneficial effects of the water-retaining polymer were observed under water deficit conditions. In plum, *Prunus salicina* trees, the application of water-retaining polymers increased fruit yield and water use efficiency of plants under water deficit [17]. Alshallash et al. (2022) [18] highlight that the application of water-retaining polymers increases water retention in the soil, causing a positive impact on the yield and quality of mango fruit (*Mangifera indica* L.) cv. Shelly in arid conditions.

In the implantation of the sour passion fruit orchard, in addition to irrigation, one of the factors that also contributes to the development of the plants is the preparation of the soil. Efficient soil management techniques, such as preparation of pits, must be implemented to promote ideal conditions for root growth and better yields of the crop [19,20]. Adequate pit volume and soil structural quality provide a better soil–water–plant relation, optimizing the efficient use of water by plants [21,22].

Thus, the objective of this study was to evaluate the chlorophyll index, gas exchange, and the yield of the sour passion fruit cv. BRS GA1 as a function of irrigation depths, pit volumes, and doses of water-retaining polymer.

## 2. Materials and Methods

### 2.1. Characterization of the Experimental Area

The experiment was conducted from September 2018 to January 2020, at the Macaquinhos Farm, located in the municipality of Remígio, in the State of Paraíba, Brazil. The municipality is georeferenced by geographic coordinates: 7°00′1.95″ latitude South, 35°47′55″ longitude West of the Greenwich Meridian and altitude of 562 m, physiographically inserted in the ‘Agreste Paraibano Mesoregion’ and ‘Curimataú Microregion’ western. The climate of the region, according to the Köppen’s classification [23], was classified as As’, which means hot and humid summer and rainy season concentrated between March and June.

Daily rainfall and evaporation data were recorded through readings in a rain gauge and Class “A” tank, respectively, and the values of temperature and relative air humidity were obtained with a Datalogger, model HT-70, installed close to the experiment (Figure 1a,b).

The soil in the experimental area was classified as Entisols Psamment [24]. Before the implementation of the experiment, six soil samples were collected per block in the 0–0.40 m layer, for the purpose of physical and chemical characterization regarding soil fertility (Table 1), according to methodologies described in [25].

### 2.2. Experimental Design and Plant Material

The experiment was carried out in randomized blocks and split plot 2 × (2 × 5), with four replications and three plants per plot (Figure 2), in which the main plot was related to irrigation depths of 70 and 100% of the crop evapotranspiration (ETc), the subplots to pit volumes of 64 dm^3^ (traditional) and 128 dm^3^ (proposed), and doses of 0; 0.5, 1.0, 1.5, and 2.0 g of water-retaining polymer per dm^3^ of soil.

Pit volumes were based on [19] and the doses of the water-retaining polymer according to the methodology contained in [5]. The water-retaining polymer used was Hydroplan^®^-EB/HyA. The plant material under study was the sour passion fruit (*Passiflora edulis* Sims), commercial cv. cv. BRS GA1, earlier known as Gigante Amarelo, propagated via seeds and obtained from a commercial nursery accredited by Embrapa and the Brazilian Ministry of Agriculture, Livestock and Supply (MAPA). The seedlings were transplanted when they presented 35 cm in height, 4 mm in diameter, and four pairs of fully expanded leaves.

### 2.3. Conducting the Experiment

The plants were distributed at a spacing of 2 m between plants and 3 m between rows in pits 0.4 m in depth with diameters of 0.45 and 0.64 m, corresponding to volumes of 64 dm^3^ and 128 dm^3^, respectively, to which 0.26% of cattle manure was added before transplanting. The amount of cattle manure applied was to raise the initial soil organic matter content from 0.4 to 2.0%. Before application, the cattle manure was characterized according to the pH and the contents of nutrients [25], as shown in Table 2.

In the respective pits of 64 dm^3^ and 128 dm^3^, 50 and 100 g of FTE-BR12 (3.9% S, 1.8% B, 0.85% Cu, 2.0% Mn and 9.0% Zn), 45 and 90 g pit^−1^ of dolomitic limestone (PRNT = 80% and 28% CaO) to raise the base saturation of the soil from 48% to 70% [26] and 9 and 18 g of potassium chloride (KCl—60% of K_2_O) to raise the potassium content of the soil from 60 to 90 mg dm^−3^.

At 30 days after transplanting (DAT), cover fertilization with nitrogen (urea, 45% N) and potassium (KCl) were applied monthly, applying 15 g and 20 g, respectively, during the transplanting period in the vegetative growth phase. During the flowering phase, the values were increased to 24 g and 30 g, and at the end of flowering, which comprises the end of the harvest, applications of 33 g and 60 g were carried out, totaling 231 and 350 g per year^−1^, respectively, of urea and potassium chloride (Figure 3).

Fertilization with phosphorus was carried out from 60 DAT, starting with the joint application of N and K, with simple superphosphate (18% P_2_O_5_, 16% Ca and 8% S), applying 50 g every three months and 100 g at the end of the fruit harvest, totaling 250 g plant^−1^ year^−1^ in 4 applications of simple superphosphate, as recommended by [27].

Irrigations were carried out with water of low salinity (without any restrictions of use) for agriculture (ECw = 0.5 dS m^−1^ and SAR = 2.2 (mmol L^−1^)^1/2^), using the micro-sprinkler irrigation system, with one emitter per plant and a flow of 60 L·h^−1^, with a pressure of 0.2 MPa. Application of the daily volume (100% of ETc) was carried out based on the crop evapotranspiration (ETc) through ET_0_ and the cultivation coefficient (Kc) of the sour passion fruit, with values of 0.43 in the vegetative phase, 0.94 in the flower in phase, and 1.04 in the fruiting phase, as recommended by [28]. The water depth equivalent to 70% of the ETc of the sour passion fruit was obtained through the product of the daily evapotranspiration requirement (100% of ETc) by a factor of 0.7.

### 2.4. Traits Analyzed

At 120 DAT, when the sour passion fruit plant was in full flower bud emission, the third pair of leaves from the branches located in the middle third of the plants was selected for reading of the chlorophyll index, chlorophyll fluorescence, and gas exchange. The evaluation of chlorophyll a (Chl a), b (Chl b), and total (Chl total) indexes were performed in the morning, using a portable ChlorofiLOG meter from FalKer^®^.

Chlorophyll fluorescence parameters referring to initial fluorescence (F_0_), maximum fluorescence (Fm), and variable fluorescence (Fv), were measured after dark adaptation of leaves for 30 min with clamps, using a modulated Plant Efficiency Analyzer—PEA II^®^ fluorometer. Gas exchange: net photosynthesis rates (A—μmol CO_2_ m^−2^ s^−1^), stomatal conductance (gs—mol H_2_O m^−2^ s^−1^), internal CO_2_ concentration (Ci—μmol CO_2_ m^−2^ s^−1^), and leaf transpiration (E—mmol H_2_O m^−2^ s^−1^) were performed with a portable infrared carbon dioxide analyzer (IRGA), model LCpro-SD from BioScientific^®^, with the temperature set at 25 °C, irradiation of 1200 μmol m^−2^ s^−1^, and air flow of 200 mL min^−1^ [29].

From the data obtained, water productivity (WP) was calculated, relating net photosynthesis to transpiration (*A*/*E*) [(μmol CO_2_ m^−2^ s^−1^)(mmol H_2_O m^−2^ s^−1^)^−1^]. Total yield (t ha^−1^) corresponds to the 2018 to 2019 harvests and was obtained by the product of production per plant and the density of plants per hectare (1667 plants ha^−1^).

### 2.5. Statistical Analysis

The collected data were subjected to the normality of distribution test (Shapiro–Wilk test) at a probability level of 0.05. Data were submitted to analysis of variance using the F test (*p* ≤ 0.05). The values referring to irrigation depths and pit volumes were compared using the F test (*p* ≤ 0.05) and the doses of water-retaining polymer by linear (polynomial regression) and non-linear (sigmoidal and peak) models. For data analysis, the statistical software SISVAR version 5.6 [30] was used. A Pearson correlation analysis was performed between the variables and a correlation matrix was created using the “GGally” statistical package from R Studio [31].

## 3. Results

### 3.1. Chlorophyll Indices and Fluorescence

The leaf chlorophyll indices and the initial fluorescence of the sour passion fruit cv. BRS GA1 were affected by the interaction among irrigation depths, pit volume, and doses of water-retaining polymer. The irrigation depth × pit volumes interaction had a significant effect on the Fv and Fm of the sour passion fruit cv. BRS GA1, as seen below in Table 3.

The Chl a index in the leaves of the sour passion fruit cultivated in the 64 dm^3^ pits and irrigated with a water depth of 70% ETc did not fit any regression model, showing an average value of 37.63 (Figure 4a). In the same pit volume, the chlorophyll a index of plants irrigated with 100% ETc in relation to those with 70% of ETc was higher in the absence and at a dose of 1.5 g of the water-retaining polymer. In the pits of 128 dm^3^, the plants irrigated with a depth of 70% of ETc presented the highest values of Chl a (41.25) in the absence of the polymer (Figure 4b). However, in plants irrigated with 100% ETc, the Chl a increased linearly from 37.15 to 45.97 with increments of the dose from 0 to 2.0 g dm^−3^ of soil water-retaining polymer, representing an increase of 23.74%.

In Chl b (Figure 4c,d), the indices were elevated with increasing doses of the water-retaining polymer from 0 to 2.0 g dm^−3^ of soil, except for plants grown in a pit of 128 dm^3^ and irrigated with water depths of 70% of ETc, which showed an average value of 11.28. In the sour passion fruit cultivated in the 64 dm^3^ pit irrigated with a 70% ETc depth, the Chl b leaf index increased from 11.58 to 14.14 when compared to the absence of the polymer for the dose of 2.0 g dm^−3^; at 100% ETc depth in the same pit volume, the highest Chl b index was obtained in the absence of the polymer (12.48), with subsequent reduction. In the pit volume of 128 dm^3^ and ETc depth of 100%, the Chl b index went from 10.80 to 14.76 between the lowest and highest doses of polymer, representing an increase of 36.66%. The association of the largest pit volume and polymer dose promoted a higher chlorophyll b index when the sour passion fruit plant was irrigated with a water depth of 100% ETc.

For the Chl total index, the plants grown in the 64 dm^3^ pit irrigated with a depth of 70% ETc, there was no regression adjustment, represented by a mean of 49.64 (Figure 4e). In plants grown in the same pit volume, under irrigation with 100% ETc, the Chl total index did not vary between treatments without and with the highest dose of the polymer, obtaining an estimated value of 56.00. In plants grown in the 128 dm^3^ pit and irrigated with 70% ETc (Figure 4f), it can be seen that increasing the polymer doses up to 1.12 g dm^−3^ provided the highest Chl total index (53.64), and in plants irrigated with 100% ETc, it increased from 48.92 to 59.67 between the pits without and with 2.0 g dm^−3^.

The initial fluorescence values obtained under different doses of water-retaining polymer did not fit satisfactorily to any regression model when the sour passion fruit was cultivated in the 64 dm^3^ pit and irrigated with a depth equivalent to 70% and 100% ETc, presenting mean values of 458.38 and 455.75, respectively (Figure 5a). On the other hand, passion fruit cultivated in a 128 dm^3^ pit and irrigated with a water depth of 70% ETc showed reductions in F0 from a polymer dose of 1.0 g dm^−3^ of soil (Figure 5b). The highest F0 values were recorded in the sour passion fruit cultivated in the 128 dm^3^ pit, under an irrigation depth of 100% ETc, which was increased from 400.49 (without polymer) to 500.41 at a dose of 2.0 g dm^−3^ of the polymer, an increase of 24.94%.

When evaluating the Fv of the sour passion fruit, it is observed that the pit volume of 128 dm^3^ and the irrigation depth of 70% of ETc, provided the highest value of Fv, 2007.8; however, it did not differ statistically from the plants cultivated in the pit volume of 64 dm^3^ under the depths of 70% and 100% ETc with values of 1945.2 and 1937.0, respectively. In the 128 dm^3^ pit, increasing the irrigation depth from 70% to 100% of ETc reduced Fv by 338.5 in sour passion fruit plants (Figure 5c). Behavior similar to Fv was observed in Fm (Figure 5d), in which the plants of sour passion fruit cultivated in the pit of 128 dm^3^ under irrigation with a water depth of 70% ETc presented the highest values of Fm (2442.8), but did not differ statistically from plants grown in a pit volume of 64 dm^3^ and irrigated with 70% ETc (2403.5) and 100% ETc (2392.6). The lowest Fm values (2120.4) were recorded in plants grown in pits with a volume of 128 dm^3^ and irrigated with 100% ETc.

### 3.2. Gas Exchange and Yield

The variables stomatal conductance, water productivity, and yield of the sour passion fruit cv. BRS GA1 were affected by the interaction of irrigation depths × pit volume × dose of water-retaining polymer (Table 4). The net photosynthesis was influenced by the interactions of PV × WRP and ID × WRP. The internal concentration of CO_2_ responded to the interaction of ID × PV, and leaf transpiration responded to the interaction of ID × WRP.

The increase in the dose of the water-retaining polymer up to 1.21 g dm^−3^ of soil increased the *A* of the sour passion fruit in irrigation depths of 100% ETc, with a value of 11.06 µmol CO_2_ m^−2^ s^−1^ (Figure 6a). In contrast, in plants irrigated with ETc equivalent to a depth of 70% ETc, the net photosynthesis rate was reduced from 15.19 to 11.27 µmol CO_2_ m^−2^ s^−1^ with an increase in the dose from 0 to 2.0 g of the polymer. It was also noted that the net photosynthesis rate was greater in sour passion fruit irrigated with the lowest irrigation depth until the application of a dose of 0.5 g dm^−3^ of the water-retaining polymer, showing no statistical differences with higher doses for the depths of 100% ETc (Figure 6a). In the plants grown in a pit volume of 64 dm^3^, the net photosynthesis rate presented an average value of 11.36 µmol CO_2_ m^−2^ s^−1^, which did not fit any tested regression model with increases in the doses of water-retaining polymer (Figure 6b). On the other hand, for those cultivated in a pit volume of 128 dm^3^, increasing the dose of the water-retaining polymer up to 1.03 g dm^−3^ raised the *A* of sour passion fruit from 9.64 to 13.53 µmol CO_2_ m^−2^ s^−1^.

The gs in the leaves of sour passion fruit cultivated in a pit volume of 64 dm^3^ and irrigated with water depths of 70% and 100% ETc did not fit satisfactorily any regression model, presenting mean values of 0.16 and 0.10 mmol H_2_O m^−2^ s^−1^, respectively (Figure 7a). In the pit volume of 64 dm^3^, only in the absence of the polymer and at a dose of 1.0 g dm^−3^ were there statistical differences in the stomatal conductance of sour passion fruit, with superiority in plants irrigated with 70% of ETc (Figure 7a). In the pit of 128 dm^3^ and irrigation with 70% of ETc, the gs also did not fit any regression model, showing a mean value of 0.14 mmol m^−2^ s^−1^ (Figure 7b). At the 100% ETc depth, the gs was increased up to a dose of 1.06 g of the polymer dose, obtaining greater stomatal opening with a value of 0.1406 mmol CO_2_ m^−2^ s^−1^, an increase of 180.0% in relation to the absence of the polymer (Figure 7b).

Regarding the Ci of sour passion fruit in plants grown in the 64 dm^3^ pit, a reduction of 2.20% was observed with an increase in irrigation depth. In the pit of 128 dm^3^, the increase in irrigation depth from 70% to 100% ETc provided the highest leaf Ci (246.0 µmol CO_2_ m^−2^ s^−1^), showing an increase of 6.17%; however, statistically, irrigation depths did not differ from each other. At the 70% depth, increasing the pit volume from 64 to 128 dm^3^ increased leaf Ci from 230.9 to 231.7 µmol CO_2_ m^−2^ s^−1^ (Figure 8a). The leaf transpiration of sour passion fruit cv. BRS GA1 irrigated with 70% ETc showed a mean value of 4.21 mmol H_2_O m^−2^ s^−1^ (Figure 8b). With 100% ETc, leaf *E* was increased to 4.28 mmol H_2_O m^−2^ s^−1^ when the water-retaining polymer was applied at a dose of 1.12 g per dm^−3^ of soil.

The water productivity of the sour passion fruit cultivated in a pit volume of 64 dm^3^ and irrigated with a depth of 70% ETc showed the highest PW of 4.04 [(mmol CO_2_ m^−2^ s^−1^) (mmol H_2_O m^−2^ s^−1^)^−1^] in the absence of the polymer, and reduced by 16.13% when subjected to the application of 2.0 g dm^−3^ of the water-retaining polymer. At the irrigation depth of 100% ETc, the WP data did not fit satisfactorily any mathematical model, presenting a mean value of 3.38 [(mmol CO_2_ m^−2^ s^−1^)(mmol H_2_O m^−2^ s^−1^)^−1^] (Figure 9a). The application of the doses of the water-retaining polymer up to a dose of 2.0 g dm^−3^ associated with a higher pit volume (128 dm^3^) and irrigation with a depth of 70% ETc increased the WP of the sour passion fruit up to 4.85 [(mmol CO_2_ m^−2^ s^−1^)(mmol H_2_O m^−2^ s^−1^)^−1^], showing an increase of 78.6% when compared to the absence of the polymer. In the sour passion fruit irrigated with ETc of 100% irrigation depth, the water productivity data did not fit satisfactorily any regression model and are represented by an average value of 3.38 [(mmol CO_2_ m^−2^ s^−1^) (mmol H_2_O m^−2^ s^−1^)^−1^] (Figure 9b).

The cultivation of sour passion fruit in a pit of 64 dm^3^ and under 70% ETc provided a higher yield compared to that irrigated with 100% ETc, with an accumulated value of 31.5 t ha^−1^ in plants submitted to 1.1 g of the polymer (Figure 10a). In the same pit volume, increasing doses of the water-retaining polymer reduced fruit yield in sour passion fruit, resulting in losses of 29.45% when comparing the highest and lowest doses of the polymer (Figure 10a). The accumulated yield of the sour passion fruit cultivated in a pit of 128 dm^3^ and irrigated with a depth of 70% ETc was increased up to a dose of 1.29 g polymer dm^−3^ of soil, with an estimated value of 18.8 t ha^−1^ of fruits (Figure 10b). In the sour passion fruit irrigated with an irrigation level of 100% ETc and cultivated in a pit volume of 128 dm^3^, the yield was increased from 10.38 to 22.16 t ha^−1^, representing gains of 113.48%. When comparing the maximum values of the accumulated yield obtained in each volume of the pit, it is verified that the volume of 64 dm^3^ was superior in 40.82% in relation to the accumulated yield of the sour passion fruit cultivated in the pit of 128 dm^3^ (Figure 10b).

### 3.3. Correlation Matrix

Figure 11 presents the multiple correlation analysis between the different variables of chlorophyll index, fluorescence, gas exchange, and sour passion fruit yield. According to the positive Pearson correlation values, the increase in photosynthetic rate in sour passion fruit is associated with greater stomatal opening (0.649 ***) and leaf transpiration (0.493 ***). This greater stomatal conductance contributed to higher evaporation rates of the sour passion fruit (0.644 **). The increase in leaf chlorophyll indices in sour passion fruit increases the initial fluorescence of chlorophyll a, with values of 0.295 * (Chla), 0.266 * (Chlb), and 0.322 ** (total Chl), respectively; however, it reduced the water use—Chla (−0.373 **) and Chl total (−0.307 *). The variable fluorescence showed a strong relationship with the maximum fluorescence (0.976 ***), positively influencing the increase in the other variable. The Chlb index was the only variable that influenced the yield of passion fruit, causing a positive effect on fruit production (0.268 *).

## 4. Discussion

The greater volume of the pits allows for greater root expansion and, consequently, greater use of water, and the application of the polymer associated with adequate water management promotes greater efficiency as a result of its soil conditioning action, in which it increases the capacity of soil water storage [32]. In this sense, the greater volume of the pit associated with greater water availability and greater water retention provided by the water-retaining polymer contributed to the higher chlorophyll indices in sour passion fruit (Figure 4). The linear increase observed in the Chl b index (Figure 4d) may be due to a negative adjustment of the enzymatic activity of Chl b reductase, responsible for the conversion of Chl b to Chl a [33], indicating that the photosynthetic apparatus is suffering damage in the chlorophyll fluorescence, which can be confirmed by observing the fluorescence data (Figure 5), which showed damage in the PSII. Behavior similar to our study was observed by [34] in eucalyptus (*Eucalyptus dunnii Maiden*) on the ground with a water-retaining polymer, which found increases in leaf chlorophyll levels depending on the irrigation frequency. The authors attributed to the water-retaining polymer a reduction in nutrient losses, mainly N, which is directly linked to the structuring composition of photosynthetic pigments in plants.

An increase in F0 may indicate that damage has occurred in the photosynthetic apparatus of plants, or a reduction in the ability to transfer excitation energy from the antenna to the photochemical reaction center [8,9]. According to Baker [35], an increase in F0 and a reduction in Fm are considered indicative of stress in plants, and according to the results observed (Figure 5d), the sour passion fruit cultivated in the pit of 128 dm^3^ irrigated with a water depth of 100% ETc were under stress, probably caused by excess water present in the passion fruit root zone, which was aggravated by the increase in polymer doses.

The reduction in Fv in the pit of 128 dm^3^ and irrigation with a depth of 100% ETc (Figure 5c) is related to the abiotic conditions to which the plants were submitted, in this case, the stress caused by excess water in the root zone of the crop, which caused damage to the photosynthetic apparatus, compromising the PSII [29]. The reduction in Fm in the greater volume of the pit with the increase in the irrigation depth (Figure 5d) demonstrates that high levels of water in the soil may have caused anoxia of the plant roots, initiating a stress condition, causing a reduction in the efficiency of photosystem II [36].

The increase in the net photosynthesis rate of the sour passion fruit at 70% ETc and an estimated dose of the water-retaining polymer of 0.5 g dm^−3^ of soil (Figure 6a) is related to the improvement in the maintenance of soil moisture close to the root system of the plants, provided by the retention of water in the polymer, made available by the respective water depth, increasing the availability of water and nutrients for plants, which contributes to increases in the net photosynthetic rate [4]. The highest net photosynthetic rates observed in Figure 6b are due to the fact that larger pit volumes allow for greater growth of lateral roots of the plants, providing greater absorption of irrigation water, since the highest concentrations of sour passion fruit roots are present at 0.60 cm from the stem, and the largest pit volume of the present study is 0.64 dm^3^. This provided a greater contact area of the roots with water, due to the greater soil moisture provided by the application of the water-retaining polymer that acts by reducing water losses and increasing the availability of water, which can positively affect the net photosynthetic rate by supplying electrons from the photolysis of H_2_O molecules [14,19,37].

The greater stomatal opening observed in the sour passion fruit irrigated with a depth of 70% ETc in the smallest pit volume and a water depth of 100% ETc in the pit with the largest volume with a dose of 1.06 g of the polymer (Figure 7) is related to the greater availability of water provided by the water-retaining polymer [4,13,14], that is, when water in the soil is not a limiting factor, the sap flow is high between the conducting vessels and the aerial part, stimulating stomatal opening and a greater frequency of water flow into the atmosphere [38,39].

The increase in Ci as a function of the greater irrigation depth and greater pit volume (Figure 8a) is related to the greater stomatal opening observed in (Figure 7b), and is indicative that, in fact, there was no restriction on the acquisition of CO_2_ by the crop, since the greater the stomatal opening, the greater the diffusion of CO_2_ in the substomatic chamber (Silva et al. 2015) [8]. However, in plants submitted to the lowest irrigation depth, the fixation process during the carboxylation phase was compromised, a fact observed in the present study, in which there was a reduction in the acquisition of CO_2_ in the lowest water depth. The increase in the leaf transpiration rate of the sour passion fruit irrigated with a water depth of 100% ETc and a dose of 1.12 g dm^−3^ of the water-retaining polymer (Figure 8b) is related to the increase in the larger stomatal opening (Figure 7b), which was observed by [40] in passion fruit, in which stomatal conductance interfered with the evapotranspiration loss of leaves.

The application of the polymer resulted in considerable increases in photosynthetic rates (Figure 6a), which compensated for the higher transpiration rates (Figure 8b) and determined the increase in WP (Figure 9), which was reflected in fruit yield (Figure 10). The greater volume of the pit provides greater WP due to the greater presence of roots within a radius of 0.60 cm from the stem [19], which absorb and reduce water losses by lateral flow, optimizing the water productivity of the sour passion fruit (Figure 9). And according to [41], monitoring the WP plays an important role in the process of water losses through plant transpiration during gas exchange and, consequently, in the assimilation of CO_2_.

The application of the polymer up to an estimated dose of 1.1 g dm^−3^ in the plants irrigated with 70% ETc (Figure 8a) probably reduced the losses of water and nutrients by leaching, contributing to the increase in yield of sour passion fruit, and according to [42,43,44], the polymer acts to maintain the physical and chemical properties of the soil, increasing the availability of nutrients, which is reflected in the increase in yield of the crops.

## 5. Conclusions

The interaction of irrigation depths × pit volumes × doses of water-retaining polymer influences chlorophyll indexes, initial fluorescence, gas exchange through stomatal conductance, leaf transpiration, and water productivity, with positive impacts on yield of the sour passion fruit. The irrigation depth of 70% ETc associated with 1.1 g of polymer dm^−3^ of soil is recommended, since it increased the yield of the sour passion fruit cultivated in a pit with a volume of 64 dm^3^. Doses of 2.0 g dm^−3^ of the water-retaining polymer, in a pit volume of 128 dm^3^ and associated with an irrigation depth of 100% ETc, probably cause stress in sour passion fruit plants due to excess water. As a future perspective, research should be conducted to understand the mechanism and interaction of soil–water-retaining polymer–plant in the acquisition of water and nutrients and the influence on the physiological and biochemical aspects and yield of fruit crops.

## Figures and Tables

**Figure 1 plants-13-00235-f001:**
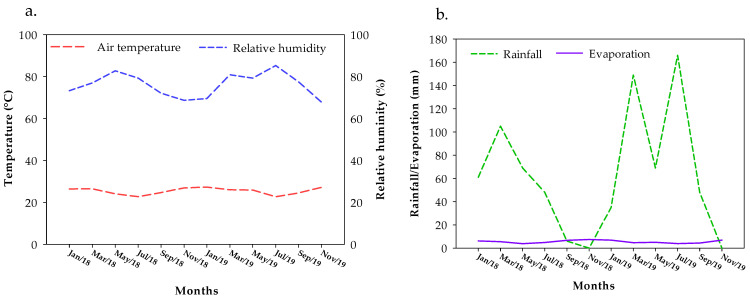
Mean monthly values of temperature and relative humidity of air (**a**) and rainfall and evaporation (**b**) during the experiment (1 January 2018 to 30 November 2019).

**Figure 2 plants-13-00235-f002:**
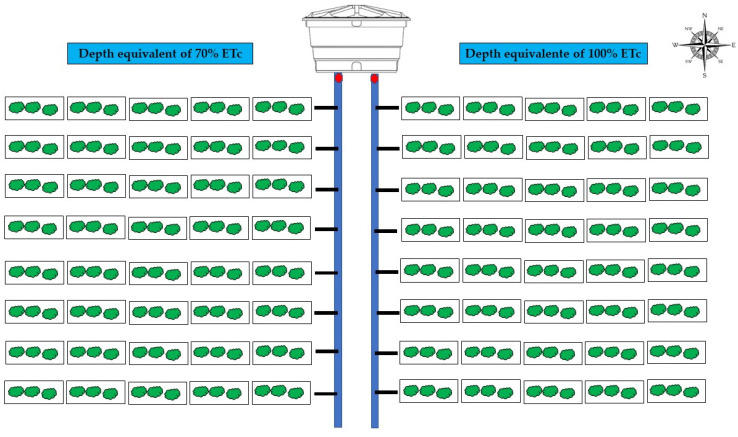
Layout of the experimental area.

**Figure 3 plants-13-00235-f003:**
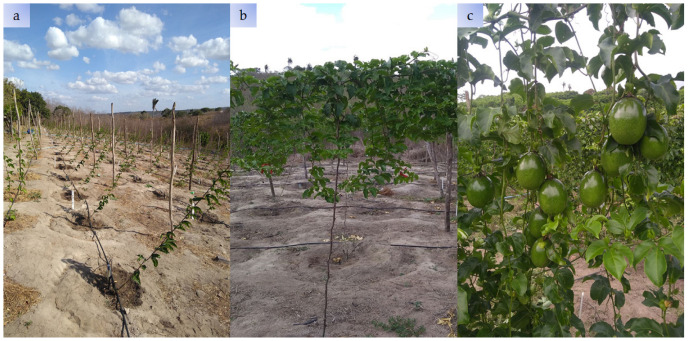
Development stages of the sour passion fruit (initial growth—vegetative (**a**), beginning of flower bud emission (**b**) and fruit production (**c**)).

**Figure 4 plants-13-00235-f004:**
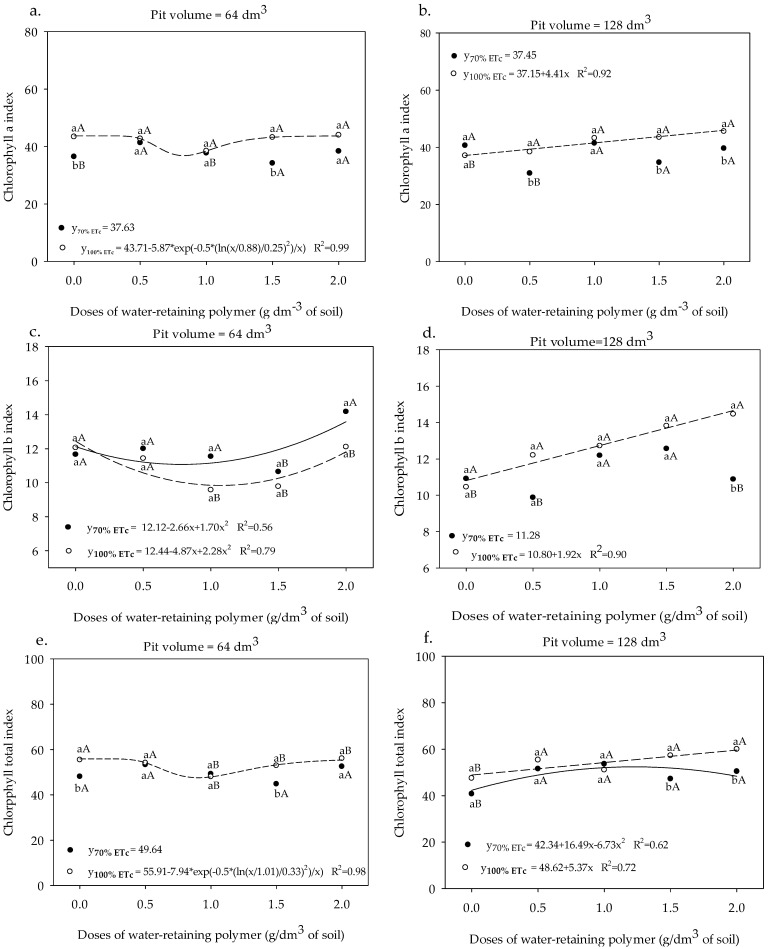
Chlorophyll a (**a**,**b**), b (**c**,**d**), and total (**e**,**f**) of sour passion fruit cv. BRS GA1 under irrigation depths, pit volumes of and, different doses of water-retaining polymer. Means with same lowercase letters do not differ in irrigation depths of 70% and 100% ETc within the same dose of the water-retaining polymer and the pit volume according to the Tukey test (*p* > 0.05), and means with same uppercase letters do not differ in pit volumes of 64 and 128 dm^3^ within each dose of water-retaining polymer and irrigation depth according to the Tukey test at 0.05 probability.

**Figure 5 plants-13-00235-f005:**
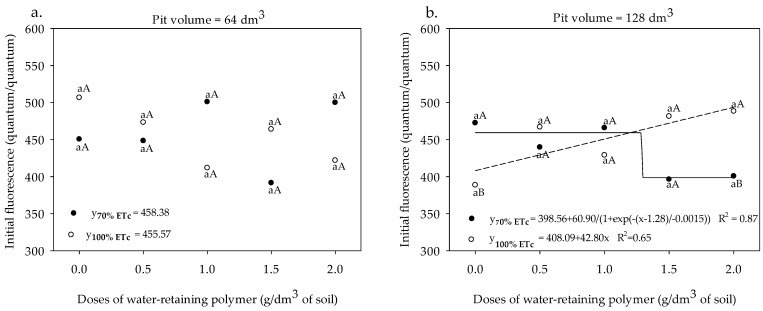
Initial fluorescence (**a**,**b**) in sour passion fruit cv. BRS GA1 under irrigation depths, pit volumes, and different doses of water-retaining polymer; variable fluorescence (**c**) and maximum fluorescence (**d**) under irrigation depths and pit volumes. Means with same lowercase letters do not differ in irrigation depths of 70% and 100% ETc within the same dose of the water-retaining polymer and pit volume according to the Tukey test at 0.05 probability, and means with same uppercase letters do not differ in pit volumes of 64 and 128 dm^3^ within each dose of water-retaining polymer and irrigation depth according to the Tukey test (*p* > 0.05). (**a**,**b**). Means with same lowercase letters do not differ in pit volumes of 64 and 128 dm^3^ within each irrigation depths according to the Tukey test at 0.05 probability, and means with equal capital letters not differ in irrigation depths of 70% and 100% within each pit volume according to the Tukey test at 0.05 probability (**c**,**d**).

**Figure 6 plants-13-00235-f006:**
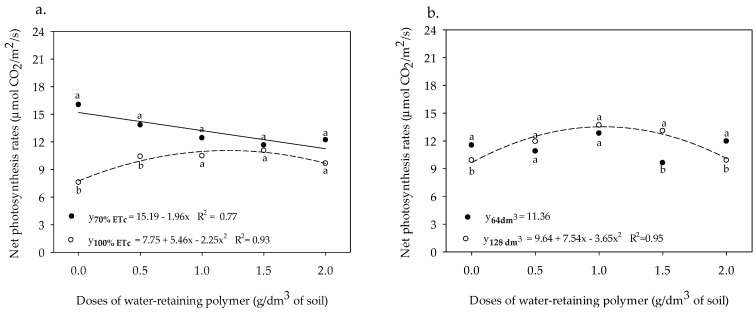
Net photosynthesis rate of sour passion fruit cv. BRS GA1 irrigated with water depths in the soil with different doses of water-retaining polymer (**a**) and cultivated in pit volumes in the soil with different doses of water-retaining polymer (**b**). Means with same lowercase letters in (**a**) do not differ in irrigation depths of 70% and 100% ETc within same dose of water-retaining polymer according to the Tukey test at 0.05 probability. Means with same lowercase letters in (**b**) do not differ in pit volumes of 64 dm^3^ and 128 dm^3^ within the same dose of water-retaining polymer according to the Tukey test at 0.05 probability.

**Figure 7 plants-13-00235-f007:**
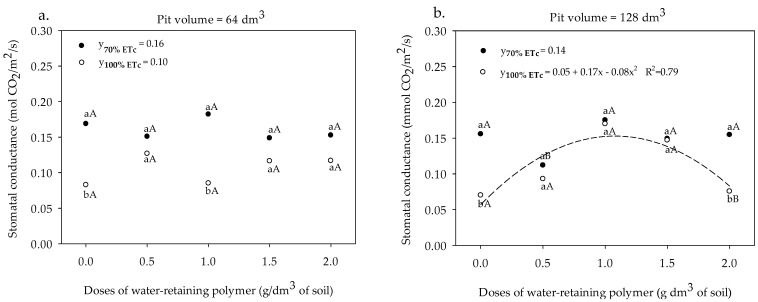
Stomatal conductance of the sour passion fruit cv. BRS GA1 irrigated with irrigation depths, pit volumes, and different doses of water-retaining polymer (**a**,**b**). Means with same lowercase letters do not differ in irrigation depths of 70% and 100% ETc within same dose of the water-retaining polymer and the pit volume according to the Tukey test at 0.05 probability, and means with same uppercase letters do not differ in pit volumes of 64 and 128 dm^3^ within same dose of water-retaining polymer and irrigation depth according to the Tukey test at 0.05 probability.

**Figure 8 plants-13-00235-f008:**
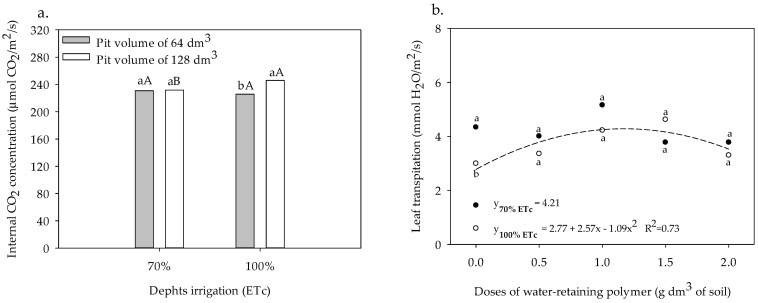
Internal CO_2_ concentration in leaves of the sour passion fruit cv. BRS GA1 cultivated in pit volumes and irrigated with irrigation depths (**a**) and leaf transpiration of sour passion fruit cv. BRS GA1 irrigated with irrigation depths and different doses of water-retaining polymer (**b**). Means with the same lowercase letters do not differ in pit volumes of 64 and 128 dm^3^ within each irrigation depth according to the Tukey test at 0.05 probability, and means with the same capital letters not differ in irrigation depths of 70% and 100% within each pit volume according to the Tukey test at 0.05 probability (**a**). Means with same lowercase letters do not differ in irrigation depths of 70% and 100% ETc within same dose of water-retaining polymer according to the Tukey test at 0.05 probability (**b**).

**Figure 9 plants-13-00235-f009:**
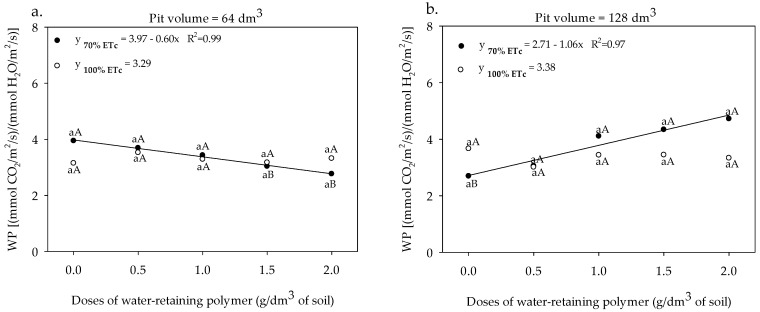
Water productivity (WP) of sour passion fruit under irrigation depth, pit volumes and, different doses of water-retaining polymer (**a**,**b**). Means with same lowercase letters do not differ in irrigation depths of 70% and 100% ETc within same dose of the water-retaining polymer and the pit volume according to the Tukey test at 0.05 probability and means with same uppercase letters not differ in pit volumes of 64 and 128 dm^3^ within same dose of water-retaining polymer and irrigation depth according to the Tukey test at 0.05 probability.

**Figure 10 plants-13-00235-f010:**
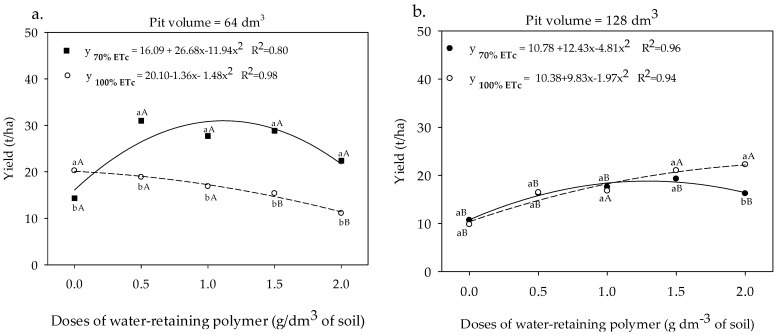
Yield of sour passion fruit under depths of irrigation, pit volumes, and different doses of water-retaining polymer (**a**,**b**). Means with same lowercase letters do not differ in irrigation depths of 70% and 100% ETc within same dose of the water-retaining polymer and the pit volume according to the Tukey test at 0.05 probability, and means with same uppercase letters do not differ in pit volumes of 64 and 128 dm^3^ within same dose of water-retaining polymer and irrigation depth according to the Tukey test at 0.05 probability.

**Figure 11 plants-13-00235-f011:**
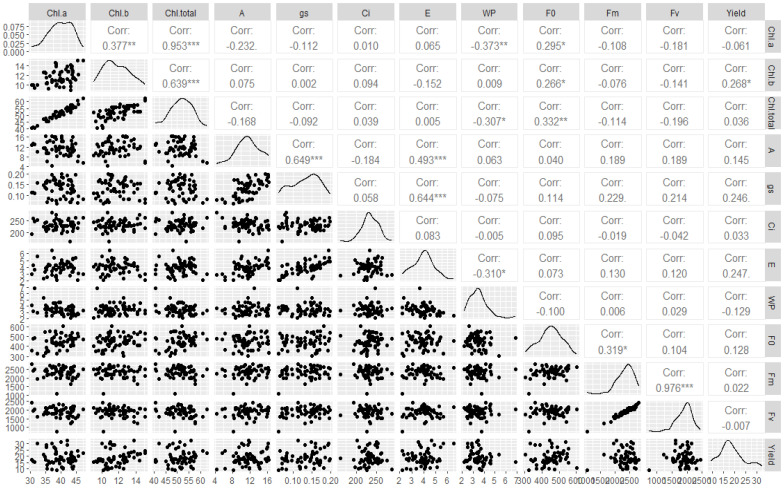
Pearson correlation matrix for the variables of chlorophyll index, fluorescence, gas exchange, and yield of the sour passion fruit under irrigation depths, pit volumes, and doses of water-retaining polymer. Chl a = chlorophyll a, Chl b = chlorophyll b, Chl total = chlorophyll total, A = net photosynthesis, gs = stomatal conductance, Ci = stomatal conductance, E = foliar transpiration, WP = water productivity, F0 = initial fluorescence, Fm = maximum fluorescence, and Fv = variable fluorescence, *, **, *** = correlation significant at 0.05, 0.01 and 0.001, respectively.

**Table 1 plants-13-00235-t001:** Physical and chemical attributes of soil fertility in the depth of 0-0.40 m in the experimental area before passion fruit cultivation.

Physical Attributes		Chemical Attributes	
Coarse Sand (g kg^−1^)	602	pH in water (1:2.5)	5.37
Fine Sand (g kg^−1^)	194	SOM (g dm^−3^)	4
Silt (g kg^−1^)	132	P–Rem (mg dm^−3^)	45.6
Clay (g kg^−1^)	72	P (mg dm^−3^)	5.5
Dispersed clay (g kg^−1^)	10.5	S (mg dm^−3^)	6.55
Bulk density (kg dm^−3^)	1.54	K^+^ (cmol_c_ dm^−3^)	0.13
Particle density (kg dm^−3^)	2.77	Ca^2+^ (cmol_c_ dm^−3^)	1.18
Total porosity (%)	44.5	Mg^2+^(cmol_c_ dm^−3^)	0.37
Degree of flocculation (%)	86	Na^+^ (cmol_c_ dm^−3^)	Dash
Dispersion index (%)	14	SB (cmol_c_ dm^−3^)	1.68
Flocculation/dispersion ratio	7.75	H^+^ + Al^3+^ (cmol_c_ dm^−3^)	1.8
Field capacity (%-v) *	8	Al^3+^ (cmol_c_ dm^−3^)	0.18
Wilting point (%-v) **	3.75	CEC (cmol_c_ dm^−3^)	3.48
Available water (%-v)	4.25	V (%)	47.85
Textural class	Sandy loam	Fertility class	Eutrophic

* Water retention at 0.01 MPa; ** water retention at 1.50 MPa; SOM = soil organic matter; SB = sum of exchangeable soil bases (SB = K^+^ + Ca^2+^ + Mg^2+^); CEC = cation exchange capacity of soil (SB + (H^+^ + Al^3+^); V(%) = saturation by exchangeable soil bases [(SB/CEC) × 100]. Extractors: SOM—Walkley-Black; P-Rem = Remaining phosphorus—CaCl_2_—0.01 mol L^−1^; P, K^+^, Na^+^—Mehlich 1; Ca^2+^, Mg^2+^, e Al^3+^—KCl—1 mol L^−1^; S = Sulfur—Acetic Acid—2 mol L^−1^; H^+^ + Al^3+^= potential acidity—calcium acetate 0.5 mol L^−1^—pH 7.0.

**Table 2 plants-13-00235-t002:** Chemical characterization of cattle manure used in the experiment.

Attribute	Value	Attribute	Value
pH (H_2_O)	8.81	Mg^2+^ (g kg^−1^)	5.0
C (g kg^−1^)	159.0	S (g kg^−1^)	1.8
C/N	19:1	B (mg kg^−1^)	21.3
Na^+^ (mg kg^−1^)	790	Cu (mg kg^−1^)	8.0
N (g kg^−1^)	8.3	Fe (mg kg^−1^)	991
P (g kg^−1^)	2.8	Mn (mg kg^−1^)	250
K (g kg^−1^)	10.4	Zn (mg kg^−1^)	58
Ca^2+^ (g kg^−1^)	8.2		

C = carbon oxidized by potassium dichromate and determined by colorimetry; C/N = carbon/nitrogen ratio; B and Fe—spectrophotometer UV-vis at the wavelengths of 460 and 508 nm, respectively; N—Kjeldahl by wet digestion; P—Mehlich-1 and photocolorimeter, 660 nm; S—spectrophotometry at 420 nm; K^+^ and Na^+^—flame photometer; Ca^2+^ and Mg^2+^—atomic absorption spectrophotometer at 422.7 and 285.2 nm, respectively; Cu—atomic absorption spectrophotometer at 324.7 nm; Mn and Zn—atomic absorption spectrometry at 231.9 and 279.5 nm, respectively, with flame-acetylene air.

**Table 3 plants-13-00235-t003:** Summary of variance analysis for leaf chlorophyll indices and chlorophyll fluorescence of the sour passion fruit plants as a function of irrigation depths (ID), pit volume (PV), and doses of water-retaining polymer (WRP).

SV	DF	Mean Squares					
		Chl a	Chl b	Chl Total	F_0_	Fm	Fv
Blocks	2	3.09 ^ns^	0.11 ^ns^	4.37 ^ns^	13,952.06 ^ns^	384,103.26 ^ns^	378,388.86 ^ns^
Irrigation depths (ID)	1	293.22 *	0.71 ^ns^	322.75 *	640.26 ^ns^	417,333.60 ^ns^	449,800.41 ^ns^
Error 1	2	4.64	0.42	0.003	17,617.86	63,411.80	49,449.26
Pit volume (PV)	1	3.66 ^ns^	3.89 *	5.50	2912.06 ^ns^	203,933.40 ^ns^	157,388.81 ^ns^
Water-retaining polymer (WRP)	4	18.09 **	5.28 **	41.03 **	1069.93 ^ns^	42,117.66 ^ns^	33,767.85 ^ns^
ID × PV	1	1.47 ^ns^	22.81 **	12.69 *	1288.06 ^ns^	363,793.06 *	409,200.41 *
ID × WRP	4	49.89 **	1.24 ^ns^	63.32 **	8125.26 *	50,025.68 ^ns^	38,505.37 ^ns^
PV × WRP	4	22.62 **	9.95 **	47.32 **	1398.40 ^ns^	31,285.06 ^ns^	31,029.52 ^ns^
ID × PV × WRP	4	44.44 **	4.04 **	64.84 **	9042.06 *	94,404.98 ^ns^	74,372.04 ^ns^
Error 2	36	1.67	0.84	3.02	3010.46	71,006.90	62,160.80
Total	59						
CV_1_		5.42	5.56	4.56	29.50	10.76	11.77
CV_2_		3.25	7.83	3.37	12.19	11.39	13.19
Mean		39.75	11.75	51.51	449,93	2339.83	1889.78

SV = sources variation; DF = degree of freedom; CV_1_ = coefficient of variation of the main plot; Chl a = chlorophyll a index; Chl b = chlorophyll b index; Chl Total = chlorophyll total index; F_0_ = initial fluorescence; Fm = maximum fluorescence; Fv = variable fluorescence; CV_2_ = coefficient of variation of the subplots; *, **, ^ns^ = significant at 0.05, 0.01 probability and not significant, respectively, by the F test at 0.05 probability.

**Table 4 plants-13-00235-t004:** Summary of analysis of variance for gas exchange and yield of the sour passion fruit plants as a function of irrigation depths (ID), pit volume (PV), and doses of water-retaining polymer (WRP).

SV	DF	Mean Squares					
		A	Gs	Ci	E	WP	Yield
Blocks	2	2.86 ^ns^	0.0011 **	2908.71 **	0.0001 ^ns^	5.00 ^ns^	194.40 **
Irrigation depths (ID)	1	180.26 *	0.0325 **	317.40	6.01 *	1.66 ^ns^	1.61 ^ns^
Error 1	2	1.86	0.0001	1.55	0.06	0.46	0.65
Pit volumes (PV)	1	1.86 ^ns^	0.0001 ^ns^	1643.26 *	2.81 *	0.60 ^ns^	248.06 **
Water-retaining polymer (WRP)	4	12.73 **	0.0010 ^ns^	605.94 ^ns^	1.77 *	0.62 ^ns^	107.80 **
ID × PV	1	0.06 ^ns^	0.0010 ^ns^	1401.66 *	0.01 ^ns^	0.06 ^ns^	345.60 **
ID × WRP	4	17.85 **	0.0063 **	504.60 ^ns^	2.89 **	0.70 ^ns^	38.85 **
PV × WRP	4	14.50 **	0.0005 ^ns^	348.97 ^ns^	0.27 ^ns^	0.55 ^ns^	59.44 **
ID × PV × WRP	4	2.15 ^ns^	0.0015 **	255.54 ^ns^	1.39 ^ns^	1.77 *	68.39 **
Error 2	36	2.29	0.0003	290.52	0.45	0.58	2.39
Total	59						
CV_1_		11.85	2.48	0.53	6.54	19.52	4.34
CV_2_		13.13	15.10	7.30	17.16	21.86	8.33
Mean		11.53	0.131	233.63	3.95	3.50	18.56

SV = sources variation; DF = degree of freedom; CV_1_ = coefficient of variation of the main plot; A = net photosynthesis rates; gs = stomatal conductance; Ci = internal CO_2_ concentration; E = leaf transpiration; WP = water productivity; CV_2_ = coefficient of variation of the subplots; *, **, ^ns^ = significant at 0.05, 0.01 probability and not significant, respectively, according to the F test at 0.05 probability.

## Data Availability

Data are contained within the article.

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
