# Peer review of "Water-Retaining Polymer and Planting Pit Size on Chlorophyll Index, Gas Exchange and Yield of Sour Passion Fruit with Deficit Irrigation"

_plants, 2024, doi:10.3390/plants13020235_

Round 1

Reviewer 1 Report

Comments and Suggestions for Authors

The article seems to be an interesting study of the sour passion fruit in different water availability controlled by the water-retaining polymers. The study indicated that plant biometric parameters can be useful in determining plant water stress. Nevertheless, some parts of the manuscript could clarified and improved before publication.

1) The aim of the study should be more precise because the literature review indicated that the implementation of water-retaining polymers in the soil is proper for the development of the passion fruit. 

2) In the methods part (section 2.1) the description of the experiments should be supplemented by the figure or scheme. The picture regarding the experimental set is more readable than the description itself. So I suggest including a graph in combination with the experiment description.

3) The term "water depth of 70% (100%) of ETc" could be clearly described in the methods. 

4) The subsection 3.3. The correlation matrix requires of extended description. Section 3.3 in its present form seems to be unnecessary. In the context of the discussion of the results, there are no important pieces of information.

5) The conclusions could be more related to the formulated aim of the study. There are lack of information about the biometric parameters relating to passion fruit cultivation.

Author Response

Impact of water-retaining polymer and planting pit-size on

chlorophyll, gas exchange and yield of sour passion fruit

with deficit irrigation – Ref.: 2753234

Cover letter

Dear Editor,

Thank you for forwarding Reviewers reports and your comments on manuscript 2753234 at present under consideration. Herewith, I am sending response to the issues raised by the Reviewers and revised manuscript with all the alterations highlighted in red color.

Review 1

1) The aim of the study should be more precise because the literature review indicated that the implementation of water-retaining polymers in the soil is proper for the development of the passion fruit.

Answer: Due modifications have been made in Introduction/objectives of study showing role of water-retaining polymers in the soil.

2) In the methods part (section 2.1) the description of the experiments should be supplemented by the figure or scheme. The picture regarding the experimental set is more readable than the description itself. So I suggest including a graph in combination with the experiment description.

Answer: OK. Correction accepted. Figures 2 and 3 were inserted.

3) The term "water depth of 70% (100%) of ETc" could be clearly described in the methods.

Answer: OK. Correction made. “The water depth equivalent to 70% of the ETc of the sour passion fruit was obtained through the product of the daily evapotranspiration requirement (100% of ETc) by the factor 0.7.”

4) The subsection 3.3. The correlation matrix requires of extended description. Section 3.3 in its present form seems to be unnecessary. In the context of the discussion of the results, there are no important pieces of information.

Answer: OK. Suggestion accepted. “According to the positive Perason correlation values, the increase in photosynthetic rate in sour passion fruit is associated with greater stomatal opening (0.649***) and leaf transpiration (0.493***). This greater stomatal conductance contributed to higher evaporation rates of the sour passion fruit (0.644**). The increase in leaf chlorophyll indices in sour passion fruit increases the initial fluorescence of chlorophyll a, with values of 0.295* (Chl a), 0.266* (Chl b) and 0.322** ( Chl total), respectively, however, it reduced the water use – Chl a (-0.373**) and Chl total (-0.307*). The variable fluorescence showed a strong relationship with the maximum fluorescence (0.976***), positively influencing the increase in the other variables. The Chl b index was the only variable that influenced the yield of passion fruit, causing a positive effect on fruit production (0.268*).“

5) The conclusions could be more related to the formulated aim of the study. There are lack of information about the biometric parameters relating to passion fruit cultivation.

Answer: Thank you for the suggestion, but we inform you that unfortunately we cannot insert the biometric variables, as they were inserted in another manuscript being evaluated in another journal. Dear reviewer, we believe the conclusion: "The interaction of irrigation depths × pit volumes × doses of water-retaining polymer influences chlorophyll levels, initial fluorescence, gas exchange through stomatal conductance, leaf transpiration, and water productivity, with positive impacts on yield of the sour passion fruit." encompasses the proposed objective of the study.

Thank you for the valuable suggestions.

Sincerely,

Prof. Dr. Antônio Gustavo de Luna Souto

Reviewer 2 Report

Comments and Suggestions for Authors

Dear author,

The manuscript looks good.

Please check the attached revised copy and make my minor corrections,

Best

Author Response

Impact of water-retaining polymer and planting pit-size on

chlorophyll, gas exchange and yield of sour passion fruit

with deficit irrigation – Ref.: 2753234

Cover letter

Dear Editor,

Thank you for forwarding Reviewers reports and your comments on manuscript 2753234 at present under consideration. Herewith, I am sending response to the issues raised by the Reviewers and revised manuscript with all the alterations highlighted in red color.

Review 2

Results of water use efficiency WUE or “water productivity” (WP) didn’t mentions in abstract?

Answer: OK. Correction made.

Authors should be add more review studies about the relation between using polymer, irrigation treatments and fruits production because it’s very little!!??

Answer: OK. Suggestion accepted. “In plum trees - Prunus salicina, the application of water-retaining polymer increased fruit yield and water use efficiency of plants under water deficit (Fadlallah et al. 2022 [17]). Al-shallash et al. (2022) [18] highlight that the application of water-retaining polymer in-creases water retention in the soil, causing a positive impact on the yield and quality of mango fruit cv. Shelly (Mangifera indica L.) in arid conditions.”

  1. Fadlallah, S.G.; Abdel-Nasser, G., Aly, M.A., Harhash, M.M., ElSegieny, A.M. Impact of super absorbent polymers (hydrogels) on water use parameters of plum trees under water stress conditions. JAAR, 2022, 27, 791-803. https://doi.org/10.21608/JALEXU.2022.173510.1097  
  2. Alshallash, K.S.; Sharaf, M.; Hmdy, A.E.; Khalifa, S.M.; Abdel-Aziz, H.F.; Sharaf, A.; Ibrahim, M.T.S.; Alharbi, K.; Elkelish, A. Hydrogel improved growth and productive performance of mango trees under semi-arid condition. Gels, 2022, 8, 602. https://doi.org/10.3390/gels8100602

Layout of experiment is not found in M&M, should be clear in it the main plot and sub-plots?!!!

Answer: OK. Suggestion accepted. “The experiment was carried out in randomized blocks and split plot 2 × (2 × 5), with four replications and three plants per plot (Figure 2) , in which the main plot was related to irrigation depths of 70 and 100% of the crop evapotranspiration (ETc), the subplots to the pit volumes 64 dm3 (traditional) and 128 dm3 (proposed), and the doses of 0, 0.5, 1.0, 1.5, and 2.0 g of water-retaining polymer per dm3 of soil.”

I suggest to change “Water Use efficiency” WP in the whole article to the modern expression “.

Answer: OK. Suggestion accepted.

  1. Mansour, H.A., Jiandong, H., Hongjuan, R., Kheiry, A.N.O., Abd-Elmabod, S.K. Influence of using automatic irrigation system and organic fertilizer treatments on faba bean water productivity. International Journal of GEOMATE, 2019, 17(62), pp. 250–259

42.Mansour, H.A., Abdel-Hady, M., Eldardiry, E.I., Bralts, V.F. 2015. Performance of automatic control different localized irrigation systems and lateral lengths for: 1-Emitters clogging and maize (Zea mays L.) growth and yield. International Journal of GEOMATE, 2015, 9(2), pp. 1545–1552

Answer: Suggestion accepted and insertion of references made.

Thank you for the valuable suggestions.

Sincerely,

Prof. Dr. Antônio Gustavo de Luna Souto

Reviewer 3 Report

Comments and Suggestions for Authors

Dear Authors,

experiment was designed nice and interesting. Statistical analyses were appropriate. However, interpretation of statistical analyses and results are not good. When you have significant interaction, than you cannot study factors separately. And you did it in whole manuscript.

Detailed comments are in manuscript.

This must be improved before publishing.

Regards,

Author Response

Impact of water-retaining polymer and planting pit-size on

chlorophyll, gas exchange and yield of sour passion fruit

with deficit irrigation – Ref.: 2753234

Cover letter

Dear Editor,

Thank you for forwarding Reviewers reports and your comments on manuscript 2753234 at present under consideration. Herewith, I am sending response to the issues raised by the Reviewers and revised manuscript with all the alterations highlighted in red color.

Review 3

Pit or pot?

Answer: Pit. The experiment was conducted under field conditions

In figures You use different labels for some of these traits and You do not use abbreviations. So it is hard to connect M&M with data in figures.

Answer: Ok. Suggestion accepted. Necessary changes were made in the text.

Please write what each parameter shows.

Answer: Dear reviewer, Thank you for the suggestion. However, the physiological and yield parameters presented in the manuscript, by definition, are consolidated in the scientific community. Thus, the authors believe that there is no need to define them.

What does HV mean?

Answer: OK. Correction made. The correct one is PV (Pit volumes)

Table 3 shows significant interaction on IDxPVxWRP. Thus, You cannot talk about significant differences in whole Figure 2 (2a, 2b, 2c, 2d, 2e and 2f).

Answer: The interaction between sources of variation in a factorial experiment may be significant, as in the examples of the IDxPVxWRP interaction. In statistics we can use the analysis strategy, the unfolding of the degrees of freedom of a factor within each level of the other factor. In this way, we can study the effect of one factor within each level of the other factor. Then there is a comparison of each irrigation depth within each dose of the polymer and the pit volume and each pit volume within each dose of the polymer and irrigation depth (Figures 4a and 4b chlorophyll a index, Figures 4c and 4d chlorophyll b index, and Figures 4e and 4f chlorophyll total index). Figure 4 in the revised manuscript corresponds to Figure 2.

Table 3 showed significant interaction IDxPVxWRP for F0. Thus You cannot talk about significant differences between ID in Fig 4a and 4b.

Answer: Yes interaction was significant, therefore a comparison was made regarding effect of each irrigation depth within each dose of the polymer and the pit volume and each pit volume within each dose of the polymer and irrigation depth (Figures 5a and 5b - Initial fluorescence)

Table 3 also showed significant interaction IDxPV for Fm and Fv. Thus You cannot talk about significant difference between ID or between PV in Fig 4c and 4d.

Answer: The interaction between sources of variation in a factorial experiment may be significant, as in the examples of the IDxPV interaction. In statistics we can use the analysis strategy, the unfolding of the degrees of freedom of a factor within each level of the other factor. In this way, we can study the effects of one factor within each level of the other factor. Therefore a comparison was made regarding effect of irrigation depth within each pit volume and the contrary (Figure 5c – variable fluorescence and Figure 5d – maximum fluorescence).

Where is Figure 3?

Answer: OK. Correction made. All numbering of figure legends have been corrected, as well as throughout the text.

In M&M (Chapter 2.4) is different abbreviation.

Answer: OK. Correction accepted. Thank you, the correction has been made.

Where is Figure 3?

Answer: OK. Correction made. All numbering of figure legends have been corrected, as well as throughout the text.

Table 4 showed significant interaction IDxWRP, Thus, in this figure (Figure 4b) You cannot state that ther is significant difference between ID.

Answer: Thanks for the observation. Figure 4b (in the revised version updated to 6b) represents the interaction between PV x WRP and non-ID x WRP. As Table 4 (ANOVA) presented a response for the interaction ID x WPR and PV x WRP, the results were presented, respectively, in Figures 6a and 6b. Figures 6a and 6b were inverted to follow the order of presentation in Table 4. The interaction between sources of variation in a factorial experiment is significant, as in the examples of the ID x WRP interaction. In statistics we can use the analysis strategy, the unfolding of the degrees of freedom of a factor within each level of the other factor. In this way, we can study the effects of one factor within each level of the other factor.

Table 4 did not show significant interaction on IDxPV. Thus Figure 4a is OK and you can show significant difference between PV. But You did not wrote what lowercase letters states in Figure 4a.

Answer: OK. Correction made. Was inserted: “Means with same lowercase letters in Figure b do not differ in pit volumes of 64 and 128 dm3 under the same dose of water-retaining polymer according to the Tukey test (p>0.05).”

Please write what does lowercase letter mean.

Answer: OK. Correction made. Was inserted: “Means with same lowercase letters in Figure b do not differ in pit volumes of 64 and 128 dm3 within same dose of water-retaining polymer according to the Tukey test (p>0.05).”

Table 4 shows significant interaction IDxPV. Thus, this parameter cannot be studied separately. Neither by ID naither by PV.

Answer: Thanks for the observation. The interaction between sources of variation in a factorial experiment may be significant, as in the examples of the ID x PV interaction. In statistics we can use the analysis strategy, the unfolding of the degrees of freedom of a factor within each level of the other factor. In this way, we can study the effects of one factor within each level of the other factor.

I don't understand what you are showing in Figure 6 a and 6b. Is it net photosynthesis (A) o stomatal conductance (Ci) and foliar transpiration (E) or Stomatal conductance (gs)?

Answer: Figure 6a (In revised manuscript Figure 8a) refers to the variable “Internal CO2 concentration”, while Figure 6b (in revised version Figure 8b) refers to “Leaf transpitation”.

As Table 4 shows significant interaction IDxPVxWRP for Gs and E in Figure 6b traits cannot be studied separately and it cannot stated any significant difference separately

Answer: Thanks for the observation. We checked the statistics and leaf transpiration did not respond to the ID x PV x WRP interaction. Responding to the ID x WRP interaction, as demonstrated in Figure 8b (in earlier version 6b). While stomatal conductance responded to the ID x PV x WRP interaction, as demonstrated and discussed in Figure 8a (in earlier version 6a).

Fo figure 6a needs to be clarified which parameter is shown and than interaction needs to be checked before further statistical analyses.

Answer: Ok, Thanks. The parameter evaluated is the internal concentration of CO2, as shown in the figure title (y-axis) and shows the effect of the interaction between pit volumes x irrigation depths.

Table 4 shows significant interaction IDxPVxWRP. Thus, Figure 7a and 7b are not shown correctly, because with significant interaction, you cannot study factors separately.

Answer: The interaction between sources of variation in a factorial experiment may be significant, as in the examples of the IDxPVxWRP interaction. In statistics we can use the analysis strategy, the unfolding of the degrees of freedom of a factor within each level of the other factor. In this way, we can study the effects of one factor within each level of the other factor. Therefore a comparison of each irrigation depth within each dose of the polymer and the pit volume and each pit volume within each dose of the polymer and irrigation depth (Figures 9a and 9b - Water productivity Figure 7a and 7b in earlier manuscript).

Table 4 shows significant interaction IDxPVxWRP. Thus, Figure 8a and 8b are not shown correctly, because with significant interaction, you cannot study factors separately.

Answer: The interaction between sources of variation in a factorial experiment may be significant, as in the examples of the IDxPVxWRP interaction. In statistics we can use the analysis strategy, the unfolding of the degrees of freedom of a factor within each level of the other factor. In this way, we can study the effects of one factor within each level of the other factor. Therefore a comparison of each irrigation depth within each dose of the polymer and the pit volume and each pit volume within each dose of the polymer and irrigation depth (Figures 10a and 10b – Yield, Figure 8a and 8b in earlier version).

Thank you for the valuable suggestions.

Sincerely,

Prof. Dr. Antônio Gustavo de Luna Souto

Reviewer 4 Report

Comments and Suggestions for Authors

Line 37: Keywords water stress, pit volume, hydrogel, physiology, fruit harvest must not be Italic.

I think the title "Impact of water-retaining polymer and planting pit-size on chlorophyll, gas exchange and yield of sour passion fruit with deficit irrigation" should be corrected, because it is not clear what the impact on chlorophyll means. Is it for the mechanism of its formation, or quantity, or degradation? And this was not studied in the work. Not even the amount of chlorophyll is determined, but only index.

Where is Figure 3?

In Table 1, 2, under the figures in Figure 2, 4, 5, 6, 7, 8, the methodology does not need to be indicated, the methodology must be described in the 2. Materials and Methods consistently, and not under the figures.

I think the sentence "ns = Significant at 0.05, 0.01 probability and not significant by the F test (p>0.05) written under the tables and figures is illogically worded. Please clarify.

Lines 133-136: What is the meaning of the newly created B e Fe, 460 e 508, etc.?

Variable fluorescence and Maximum fluorescence measurement units are not indicated in Figure 4.

Lines 471, 475-476: the literature citation is duplicated by surnames and numbers.

Author Response

Impact of water-retaining polymer and planting pit-size on

chlorophyll, gas exchange and yield of sour passion fruit

with deficit irrigation – Ref.: 2753234

Cover letter

Dear Editor,

Thank you for forwarding Reviewers reports and your comments on manuscript 2753234 at present under consideration. Herewith, I am sending response to the issues raised by the Reviewers and revised manuscript with all the alterations highlighted in red color.

Review 4

Line 37: Keywords water stress, pit volume, hydrogel, physiology, fruit harvest must not be Italic.

Answer: OK. Correction accepted.

I think the title "Impact of water-retaining polymer and planting pit-size on chlorophyll, gas exchange and yield of sour passion fruit with deficit irrigation" should be corrected, because it is not clear what the impact on chlorophyll means. Is it for the mechanism of its formation, or quantity, or degradation? And this was not studied in the work. Not even the amount of chlorophyll is determined, but only index.

Answer: Thanks for the observation. Suggestion accepted. Based on the suggestion, we decided to suppress the word impact so as not to cause conflicts of understanding among readers. “Water-retaining polymer and planting pit-size on chlorophyll indexes, gas exchange and yield of sour passion fruit with deficit irrigation”.

Where is Figure 3?

Answer: OK. Correction made. All numbering of figure legends have been corrected, as well as throughout the text.

In Table 1, 2, under the figures in Figure 2, 4, 5, 6, 7, 8, the methodology does not need to be indicated, the methodology must be described in the 2. Materials and Methods consistently, and not under the figures.

Answer: Thanks for the comments. We would like to inform that at the bottom of Tables and Figures explanation is provided either for the abbreviated terms used or for the comparison of means. The details of methodologies used in conducting experiment or performing determinations are given in Materials and Methods.

I think the sentence "ns = Significant at 0.05, 0.01 probability and not significant by the F test (p>0.05) written under the tables and figures is illogically worded. Please clarify.

Answer: OK. Correction accepted. Thanks for th observation. It was fixed to: ; *, **, ns = Significant at 0.05, 0.01 probability and not significant, respectively, by the F test at  0.05 probability.

Lines 133-136: What is the meaning of the newly created B e Fe, 460 e 508, etc.?

Answer: These are the wavelength values ​​in nm for reading in spectrophotometers for each specific element, according to the methodologies recommended in Texeira et al. (2017).

Variable fluorescence and Maximum fluorescence measurement units are not indicated in Figure 4.

Answer: OK. Correction made. The measurement unit for fluorescence is quantum quantum-1.  

Lines 471, 475-476: the literature citation is duplicated by surnames and numbers.

Answer: OK. Correction made. In the citation, the names of the authors have been removed.

Thank you for the valuable suggestions.

Sincerely,

Prof. Dr. Antônio Gustavo de Luna Souto

Round 2

Reviewer 4 Report

Comments and Suggestions for Authors

Unfortunately, the authors did not understand the note and left the letters e untidy. Probably must not be e.g. B e Fe, but B and Fe, etc.

Author Response

Impact of water-retaining polymer and planting pit-size on chlorophyll, gas exchange and yield of sour passion fruit with deficit irrigation – Ref.: 2753234  Cover letter  Dear Editor, Thank you for forwarding Reviewers reports and your comments on manuscript 2753234 at present under consideration. Herewith, I am sending response to the issues raised by the Reviewers and revised manuscript with all the alterations highlighted in red color.  Review 4  Unfortunately, the authors did not understand the note and left the letters e untidy. Probably must not be e.g. B e Fe, but B and Fe, etc.  Answer:  We apologize for the lack of understanding and thank you for the correction. “C = carbon oxidized by potassium dichromate and determined by colorimetry; C/N = car-bon/nitrogen ratio; B and Fe - spectrophotometer UV-vis at the wavelengths of 460 and 508 nm, respectively; N - Kjeldahl by wet digestion; P - Mehlich-1 and photocolorimeter, 660 nm; S - spectrophotometry at 420 nm; K+ and Na+ - flame photometer; Ca2+ and Mg2+ - atomic absorption spectrophotometer at 422,7 and 285,2 nm, respectively; Cu - atomic ab-sorption spectrophotometer at 324.7 nm; Mn and Zn - atomic absorption spectrometry at 231.9 and 279.5 nm, respectively, with flame-acetylene air.”  Thank you for the valuable suggestions.  Sincerely,  Prof. Dr. Antônio Gustavo de Luna Souto Corresponding Author  
